# Biodiversity of Lactic Acid Bacteria in Traditional Fermented Foods in Yunnan Province, China, and Comparative Genomics of *Lactobacillus plantarum*

Hong Li [1,2,3], Jiang Zhu [1,2,3], Yue Xiao [1,2,3], Shiyao Zhang [1,2,3], Yuwei Sun [1,2,3], Zhijia Liu [1,2,3], Chuanqi Chu [4], Xiaosong Hu [1,2,3,5] and Junjie Yi [1,2,3,*]

1   Faculty of Food Science and Engineering, Kunming University of Science and Technology, Kunming 650500, China; lihong12581@163.com (H.L.); jiuyuan1111303@163.com (J.Z.); xyz1620@163.com (Y.X.); zs1zhangshiyao@163.com (S.Z.); sunyuweiv@163.com (Y.S.); zhijia_liu@outlook.com (Z.L.); huxiaos@263.net (X.H.)
2   Yunnan Engineering Research Center for Fruit & Vegetable Products, Kunming 650500, China
3   International Green Food Processing Research and Development Center of Kunming City, Kunming 650500, China
4   School of Food Science and Technology, Jiangnan University, Wuxi 214122, China; 7180112069@stu.jiangnan.edu.cn
5   College of Food Science and Nutritional Engineering, China Agricultural University, Beijing 100083, China
*   Correspondence: junjieyi@kust.edu.cn

**Abstract:** The diversity of lactic acid bacteria (LAB) in fermented foods in Yunnan currently lacks large-scale and systematic research. A total of 638 fermented foods were collected from 84 regions in Yunnan for diversity analyses. The results show that the dominant strains in various types of fermented foods were different. Additionally, the majority of the LAB were better adapted to regions with a temperature of 15–20 °C and a humidity of 64–74%. *Lactobacillus plantarum* (*L. plantarum*) was the most abundant of all the strains and was widely distributed in the 84 regions. Genetically, the guanine plus cytosine (GC) content of *L. plantarum* ranged from 35.60% to 47.90%, with genome sizes from 2.54 Mb to 5.76 Mb. A phylogenetic analysis revealed that the habitat source and geographic origin had little influence on the homologous genes of *L. plantarum*. The genetic diversity of *L. plantarum* was mostly represented by functional genes and carbohydrate utilization. This research provides valuable insights into the microbiota of different types of fermented foods in Yunnan. Meanwhile, a genetic diversity analysis of *L. plantarum* may help us to understand the evolutionary history of this species.

**Keywords:** biodiversity; lactic acid bacteria; fermented foods; comparative genomics; carbohydrate utilization

## 1. Introduction

The sour, spicy, salty, umami and stimulating tastes provided by fermented foods have been considered the soul of Yunnan cuisine. Among these five flavors, the sour taste mainly comes from the lactic acid produced by lactic acid bacteria (LAB). LAB form a large class of non-spore-forming, catalase-negative, Gram-positive and facultatively anaerobic bacteria that can utilize carbohydrates to produce lactic acid [1,2]. LAB play a critical role in food fermentation, and they can generate various flavor-active and bioactive compounds, conferring unique tastes and nutritional functions to food [3,4] and even prolonging the shelf life of products [5]. Moreover, LAB could be beneficial for host health, providing that they are administered in adequate amounts and are able to colonize in the gut of the host [6–8]. In conclusion, LAB play an important role in food processing and health promotion. Therefore, it is necessary to identify and investigate the LAB in fermented foods.

In recent years, some studies have begun to examine the microbial resources in fermented foods in Yunnan Province. For example, Ye et al. established that *Lactobacillus brevis* and *Candida* sp. F15 were the dominant strains in pickled chili pepper [9]. Liu et al.

isolated 260 strains of LAB from 30 traditional fermented *douchi* in six cities and counties in Yunnan [10]. Liu et al. considered the composition of the LAB in 20 acid whey samples collected from Yunnan [11]. Nevertheless, these studies mainly focus on a certain type of fermented food, and no large-scale systematic analysis has been conducted of the LAB diversity of fermented foods in Yunnan. Therefore, in this research, the diversity of the LAB in 638 samples of eight kinds of fermented foods collected from 84 regions in Yunnan Province was analyzed for the first time.

*Lactobacillus plantarum* is a nomadic species, and it commonly exists in a variety of fermented foods, including fermented vegetables, fermented meat and fermented beans [12]. Previous studies have demonstrated the existence of links between specific environmental factors and genes [13]. In addition, another study also showed extensive gene loss and lateral gene transfer in LAB during co-evolution with their habitats [14]. Thus, a certain correlation between the genetic diversity and different fermented foods of *L. plantarum* might be identified. Furthermore, with the development of molecular biology and genome sequencing technology, the study of microorganisms has gradually shifted from physiological and biochemical characteristics to the genome level [15]. An increasing number of studies have confirmed that comparative genomics provides important information for the functional analysis and determination of the evolutionary changes associated with the niche adaptation of LAB [16–18].

The aims of this research were to isolate and identify the dominant strains in eight types of fermented foods using a culture-dependent method and 16S rRNA sequencing technology and to explore the diversity of the LAB in Yunnan. Furthermore, comparative genomics was used to analyze the genetic diversity of *L. plantarum*. This research sets the stage for the further development of LAB in fermented foods in Yunnan and provides information for their application in the food industry. In addition, the genome sequencing and comparative genomics analysis may help to reveal the biotechnological potential of *L. plantarum* strains in Yunnan regions while facilitating their future development as probiotics.

## 2. Materials and Methods

### 2.1. Isolation and Identification of LAB

A total of 8 types of fermented foods were collected from 84 regions in Yunnan Province of China. Each sample was collected in a sterile 50-mL tube and then transported to a laboratory (Kunming, Yunnan, China). All samples were stored at 4 °C until further analysis. The isolation of LAB was based primarily on the method devised by Sun et al. [19]. Briefly, 0.5 g of a solid sample or 0.5 mL of a liquid sample was mixed with 4.5 mL of sterilized saline (0.85% *w/v*, NaCl) and ground uniformly with a mortar and pestle. Each sample was serially diluted with sterilized saline (0.85% *w/v*, NaCl), spread on Man–Rogosa–Sharpe (MRS) agar plates and incubated under anaerobic conditions at 37 ± 1 °C for 48 ± 2 h. Colonies with different morphologies (size, color, shape and surface) were then selected and cultured on another MRS agar plate for purification [20]. After that, the purified strain was identified by carrying out Gram staining and 16S rDNA sequencing. The 16S rDNA gene sequences were identified to the species level using the BLAST program on the NCBI website (http://:www.ncbi.nlm.nih.gov/ (accessed on 10 November 2022)). The strains identified as LAB were maintained in 20% (*v/v*) glycerol at −80 °C.

### 2.2. Diversity Analysis

#### 2.2.1. Distribution of LAB in Different Fermented Foods

In order to obtain a more direct insight into the distribution of LAB in the different fermented foods, a diversity coefficient, *D*, was introduced. The diversity coefficient *D* in the different strains was defined as

$$D = \text{(Number of certain strain)}/\text{(Number of samples containing this type of strain)} \quad (1)$$

The diversity coefficient *D* indicates the frequency of distribution of particular strains in a sample. A higher diversity coefficient indicates that the strain is the dominant strain in the sample.

2.2.2. Distribution of LAB in Different Regions of Yunnan

Recent studies have shown that climate variables (e.g., temperature and humidity) are important indicators in explaining microbial distribution and richness [21,22]. Therefore, in order to better describe the influences of temperature and humidity on the distribution of LAB in different regions of Yunnan, the average annual temperature was divided into three intervals: a low-temperature region (11–15 °C), a medium-temperature region (15–20 °C) and a high-temperature region (20–25 °C). The average annual humidity was also divided into three intervals: a low-humidity region (54–64%), a medium-humidity region (64–74%) and a high-humidity region (74–84%). Data on the annual average temperature and annual average humidity in 84 regions were obtained from China Weather (http://www.weather.com.cn/ (accessed on 12 January 2022)). FineBi (https://www.finebi.com/ (accessed on 18 January 2022)) was used for data visualization.

*2.3. Comparative Genomics Analysis*

*L. plantarum* was collected via centrifugation at 3800 rpm for 5 min and then submitted to a sequencing company (Shanghai Majorbio Bio-pharm Technology Co., Ltd., Shanghai, China) for DNA extraction and a sequencing analysis [19]. In terms of genome assembly, after removing the low-quality data obtained from the sequencing platform, the clean reads of each strain were assembled using SOAPdenovo 2.0 and SPAdes 3.11 software.

Glimmer (version = 3.02, http://ccb.jhu.edu/software/glimmer/index.shtml (accessed on 18 November 2022)) was used to predict the guanine plus cytosine (GC) and genome size content of *L. plantarum*. The calculation of the average nucleotide identity (ANI) between strains was carried out using pyani (version = 0.2.12, https://github.com/widdowquinn/pyani (accessed on 18 November 2022)). Then, TBtools (https://github.com/CJ-Chen/TBtools/releases (accessed on 18 November 2022)) was used to draw an ANI heatmap. In order to determine the degree of openness of the *L. plantarum* genome, the pan-genome and core genome were analyzed using PGAP (version = 1.2.1, https://sourceforge.net/projects/pgap/files/PGAP-1.2.1/ (accessed on 18 November 2022)), and the R language was used for data visualization. Orthomcl (version = 1.4) was used to compare the protein sequences of *L. plantarum*, and the script was used to draw a Venn diagram based on the obtained data. A phylogenetic tree was constructed based on homologous genes, and Evolview (https://www.evolgenius.info/evolview/#/ (accessed on 25 November 2022)) was used to annotate and beautify the phylogenetic tree. A functional annotation of the genome of *L. plantarum* was performed using the Clusters of Orthologous Groups of Proteins (COG, http://eggnog5.embl.de/#/app/home (accessed on 18 November 2022)) database. The E value was set to $1 \times 10^{-5}$. Ultimately, the carbohydrate-active enzymes of *L. plantarum* were predicted using the Carbohydrate-Active Enzyme (CAZY) database (http://www.cazy.org/ (accessed on 18 November 2022)).

*2.4. Statistical Analyses*

IBM SPSS Statistics 23 was used for statistical analyses. The results are expressed as the mean $\pm$ standard deviation (SD). Scheffe and Tamhane's T2 test were used to examine the differences between the means of the parameters. $p < 0.05$ was considered statistically significant.

**3. Results**

*3.1. Diversity of Distribution of LAB*

3.1.1. Species and Numbers of Culturable LAB

In total, 8 types of fermented foods were collected from 14 prefecture-level cities (84 regions) in Yunnan (Figure 1A,B). Among these fermented foods, there were 407 fermented vegetables, accounting for about 63.79% of the total sample. The next fermented samples

were beans (*n* = 126), wine (*n* = 53), meat (*n* = 18) and fruits (*n* = 16), accounting for 19.75%, 8.31%, 2.82% and 2.51% of the total sample, respectively. The lowest numbers of samples were for dairy products (*n* = 10), fermented flour products (*n* = 6) and fermented tea (*n* = 2), accounting for 1.57%, 0.94% and 0.31% of the total samples, respectively. A total of 638 fermented foods were collected from different regions in Yunnan Province, but LAB were only isolated from 66% (421) of these samples (Figure S1). Among the 421 fermented food samples, a total of 64 species and 1487 strains of LAB were isolated. The specific numbers of each strain are shown in Figure 1C. Among all the strains, the number of *L. plantarum* was the highest.

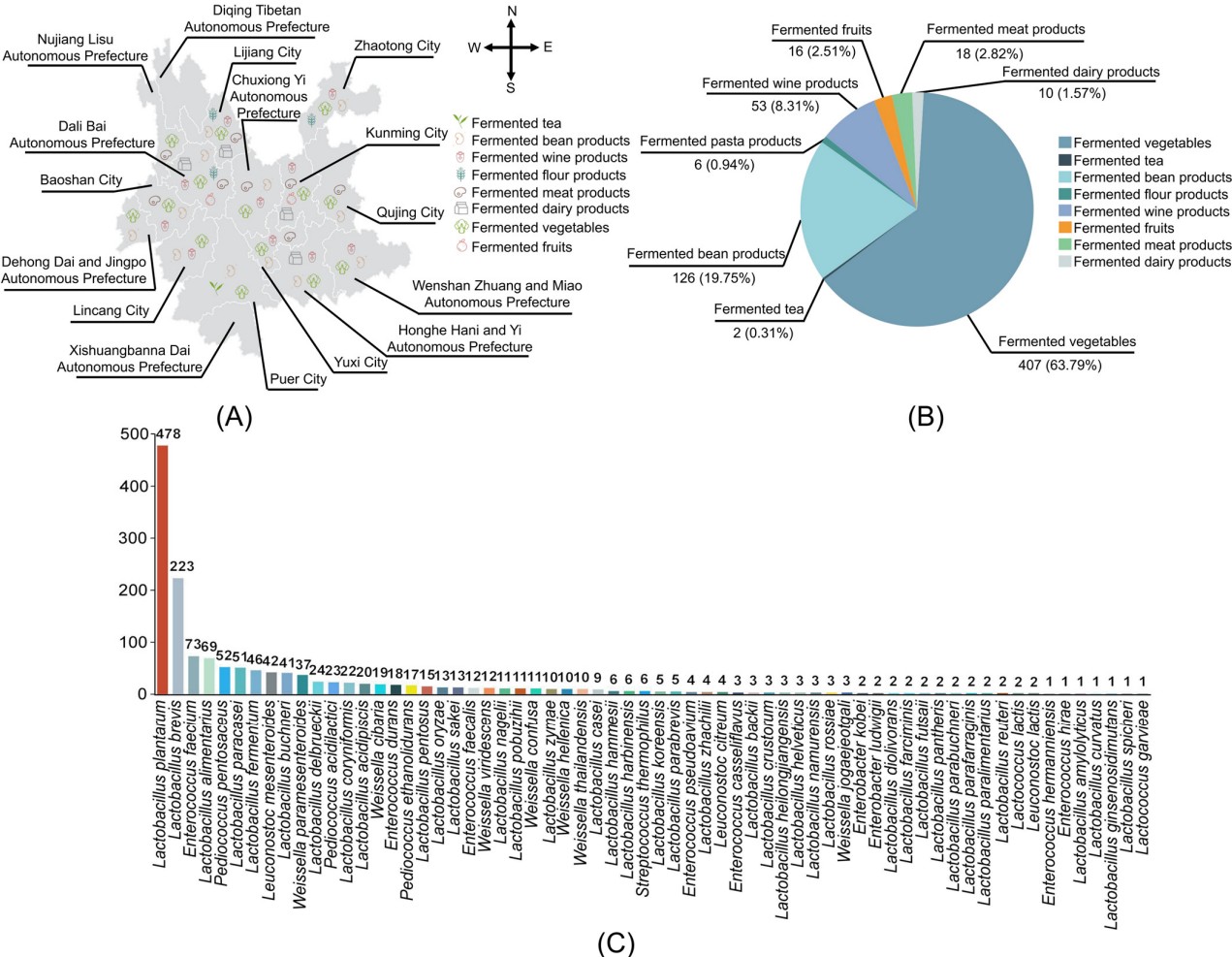

**Figure 1.** An overview of fermented foods collected from Yunnan Province. (**A**) The distribution of eight types of fermented foods in different regions of Yunnan Province; (**B**) the numbers of each type of fermented food; (**C**) the specific number of LAB.

At present, there are two main technologies used for the study of microbial diversity: culture-dependent and culture-independent technologies [23,24]. Culture-dependent technology relies on the culture medium and culture conditions. Culture-independent technology depends on advanced sequencing technologies (PCR-DGGE, metagenome, 16s full-length sequencing, etc.) [24]. The culture-dependent method is normally considered to be time-consuming and less effective, as only culturable microorganisms can be isolated, and this method provides insufficient information to study the microbial profiles in fermented foods [7]. Although it is more difficult to reflect the full details of the microbial distribution in fermented foods using the culture-based method than culture-independent techniques, the culture-dependent method is the most direct method that can be used to obtain living organisms for preservation and further study [25]. However, it should be

noted that the culture-dependent method may overlook some potential LAB in different fermented foods.

### 3.1.2. Dominant LAB in Different Types of Fermented Foods

At the genus level, a total of six genera of LAB were isolated from the fermented vegetable samples (Figure 2). Among the six genera, *Lactobacillus* was the predominant genus. As shown in Figure 3, a total of 51 species of LAB were isolated from the fermented vegetable samples. The diversity coefficient of *Leuconostoc mesenteroides* was 4, indicating that this strain was the dominant strain in the fermented vegetables. Some strains commonly found in fermented vegetables, such as *L. plantarum*, *L. brevis*, *Lactobacillus pentosus* and *Enterococcus durans*, were also isolated from the samples. Four genera and thirty-eight species of LAB were discovered in the fermented bean products (Figures 2 and 3). *Lactobacillus* was the most predominant genus isolated from the fermented bean products. Among the fermented soybean products, the highest diversity coefficient was 1.5, which was obtained for *Lactobacillus backii*, *Lactobacillus pobuzihii* and *Lactobacillus zhachilii*. Fifteen species of LAB belonging to six genera were isolated from fifty-three wine samples (Figures 2 and 3). *Lactobacillus* was the dominant genus in the wine samples. At the species level, *Weissella confuse* ($D = 4$) had the highest diversity coefficient. A total of five genera and thirteen species of LAB were isolated from the fermented meat products, with *Lactobacillus* being the dominant genus (Figure 2). At the species level, it was found that the dominant strain was *L. mesenteroides* ($D = 3.5$) (Figure 3). As shown in Figures 2 and 3, 12, 4, 6 and 2 species of LAB were isolated from the fermented dairy products, fermented fruits, fermented flour samples and fermented tea samples, respectively. In the fermented dairy products, fermented fruits and fermented flour samples, *Lactobacillus* was the predominant genus. In the fermented tea samples, *Pediococcus* was the dominant genus. The dominant LAB in the fermented dairy products, fermented fruits, fermented flour samples and fermented tea samples were *Weissella paramesenteroides* ($D = 4.5$), *Lactobacillus nagelii* ($D = 4$), *Weissella viridescens* ($D = 4$) and *Pediococcus acidilactici* ($D = 5$), respectively.

Traditional fermented foods are created by using microorganisms for fermentation, and they include various vegetables, beans, meat, cereals, milk and tea [26,27]. Traditional fermented foods have a long history, and they can be found in many variations. LAB are an important class of microorganisms that are active in the process of fermentation [28,29]. In the process of preparation and fermentation, LAB participate in the fermentation process of raw materials, acting as starter cultures or providing the fermentation environment [7]. The fermentation of vegetables, fruits and cereals relies on the LAB of raw materials as a source of inoculum [30]. Other fermentation foods, including fermented dairy products, fermented beans, fermented meat products, sourdough and fermented teas, are controlled by back-slopping or LAB in the production environment [30]. Due to the different sources and raw materials of LAB, the dominant strains among fermented foods are also different. Therefore, in this study, it was found that the dominant strains in eight types of fermented foods were different. In addition, except for the dominant genus in the fermented tea being *Pediococcus*, the dominant genus in the other seven fermented foods was *Lactobacillus* (Figure 2). Furthermore, *L. plantarum* was found in almost all types of fermented foods, except for in the fermented fruits (Figure 3), indicating that *L. plantarum* is widely distributed in fermented foods.

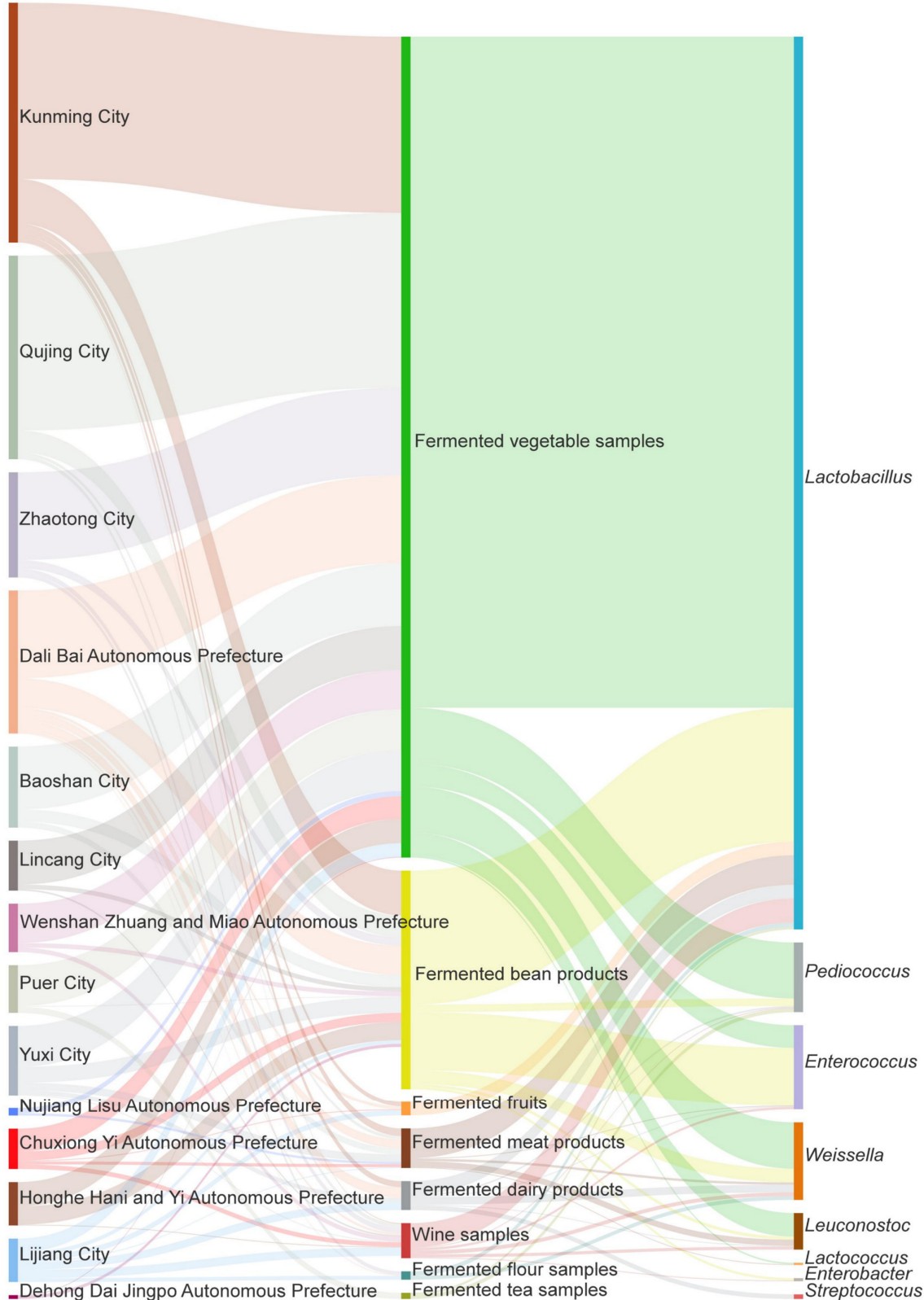

**Figure 2.** A Sankey diagram showing the relationships among regions, fermented foods and species.

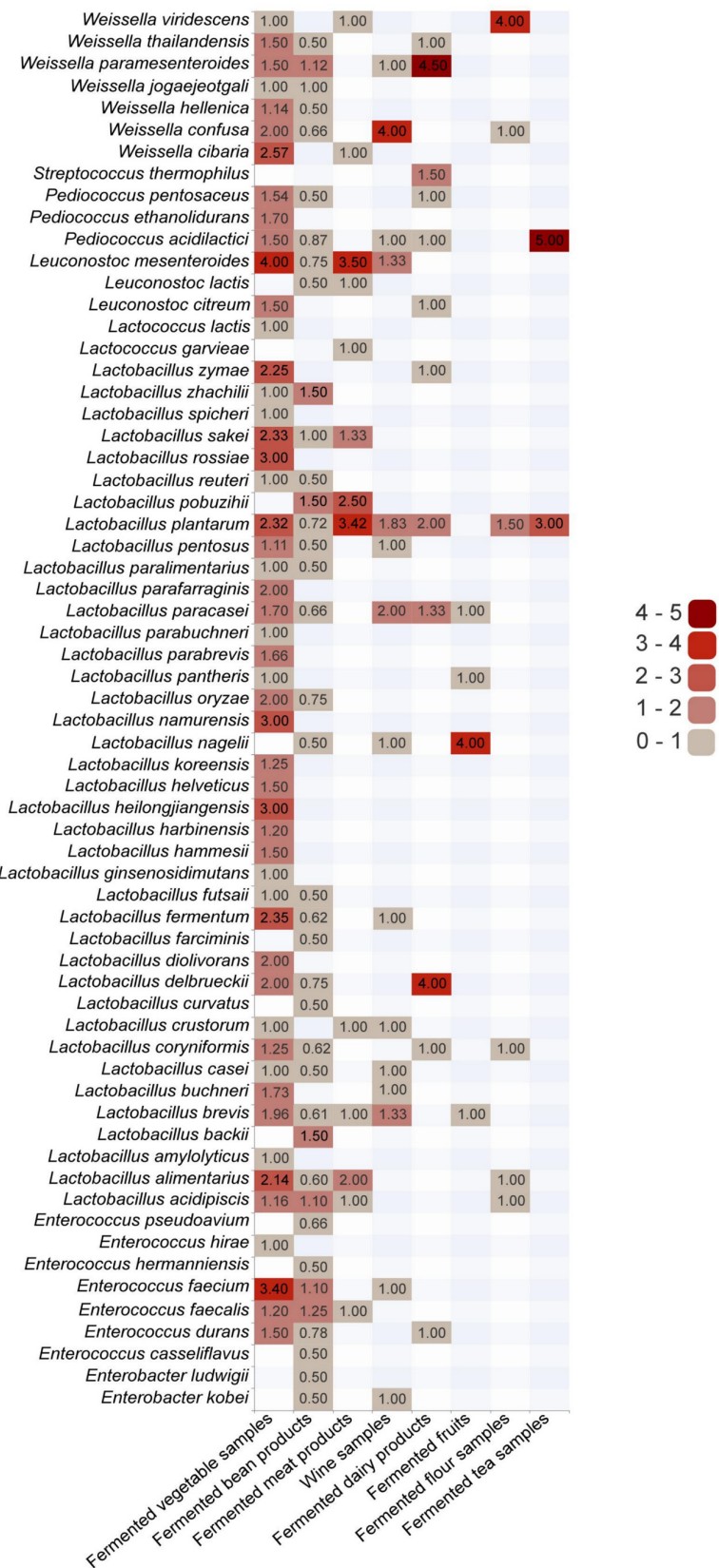

**Figure 3.** Heat map of the diversity coefficient of LAB in different fermented foods.

### 3.1.3. Distribution of LAB in Different Regions of Yunnan

The strains isolated from regions where the annual average temperature ranges from 11 to 15 °C mainly included *L. plantarum*, *Lactobacillus alimentarius*, *L. brevis*, *Lactobacillus*

*fermentum* and *L. nagelii* (Figure 4). Moreover, the strains isolated from regions where the annual average temperature ranges from 15 to 20 °C mainly included *L. plantarum, L. brevis, Enterococcus faecium, L. alimentarius, Lactobacillus buchneri, P. acidilactici, E. durans, Lactobacillus casei, Lactobacillus coryniformis, L. fermentum, L. pobuzihii, L. mesenteroides, Weissella cibaria, W. confusa, W. paramesenteroides* and *Weissella thailandensis*. Finally, the strains isolated from regions where the annual average temperature ranges from 20 to 25 °C mainly included *L. plantarum, L. brevis, L. buchneri* and *L. pobuzihii*.

The strains isolated from regions where the average annual humidity ranges from 54 to 64% mainly included *L. plantarum, L. brevis, L. nagelii* and *W. thailandensis* (Figure 4). Secondly, the strains isolated from regions where the average annual humidity ranges from 64 to 74% mainly included *L. plantarum, L. brevis, E. faecium, L. alimentarius, L. buchneri, E. durans, L. casei, L. coryniformis, L. fermentum, L. pobuzihii, L. mesenteroides, W. cibaria, W. confusa* and *W. paramesenteroides*. Finally, the strains isolated from regions where the average annual humidity ranges from 74 to 84% mainly included *L. plantarum, P. acidilactici, E. faecium, L. alimentarius, L. brevis* and *L. buchneri*.

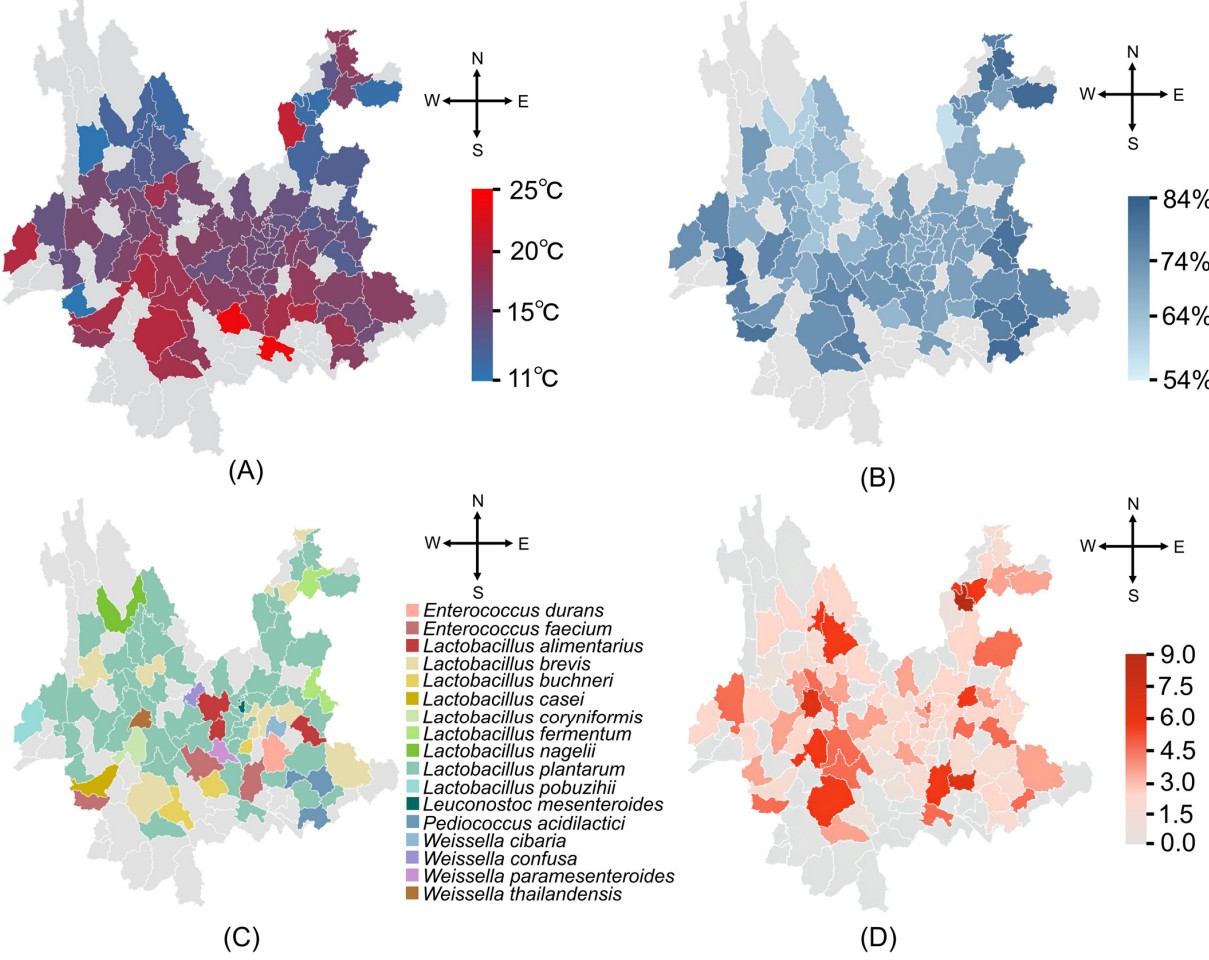

**Figure 4.** *Cont.*

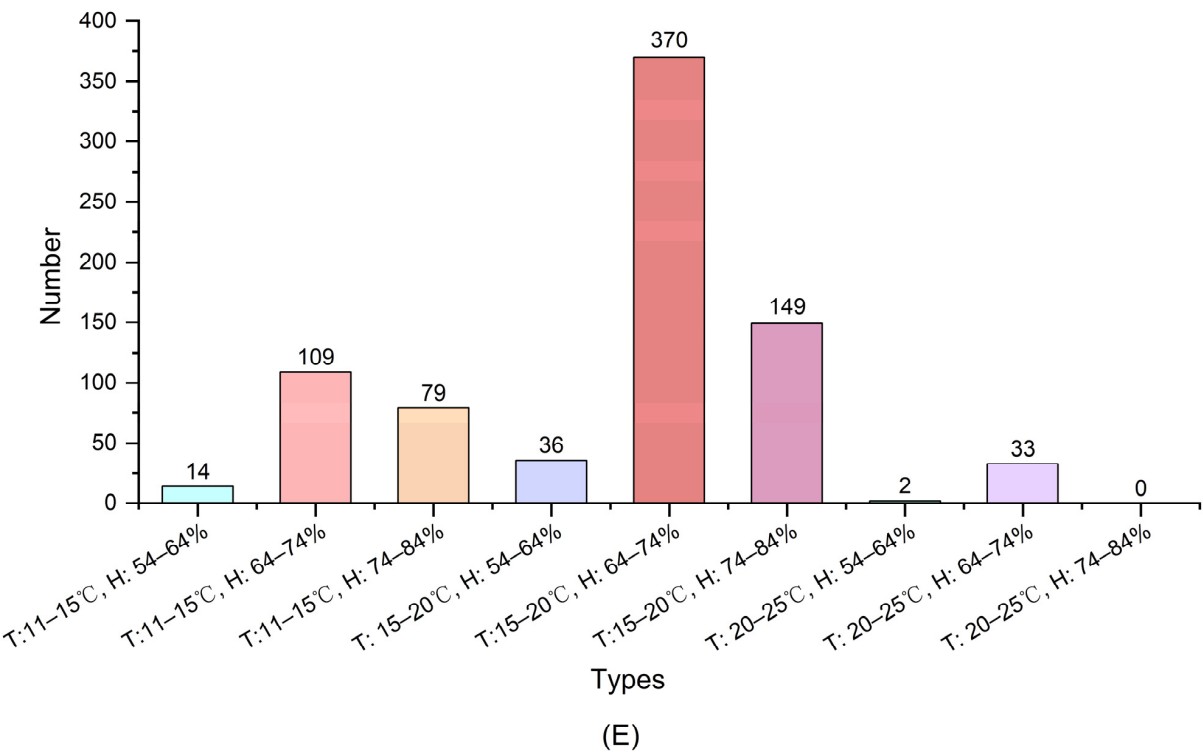

(E)

**Figure 4.** Distributions of LAB in different regions of Yunnan. (**A**) Average annual temperature in 84 regions; (**B**) average annual humidity in 84 regions; (**C**) the most frequently isolated strains in 84 regions; (**D**) heat map of the number of LAB species in 84 regions of Yunnan; (**E**) the number of LAB at different temperature and humidity. T, temperature; H, humidity.

Microorganisms typically grow and multiply at ambient temperatures lower than their optimum growth temperature [31]. The optimum growth temperature of most LAB is 37 °C [30]. In Yunnan, the annual mean temperature fluctuates within the range of 11 to 25 °C (Figure 4A), which is much lower than the optimum temperature for LAB growth. Changes in temperature affect the enzyme activity and membrane fluidity of microorganisms [32,33]. Microorganisms in warm regions have higher enzyme activities, biological activities and productivities than microorganisms in cold regions [34,35]. Additionally, with an increase in temperature, the fluidity of the cell membrane is accelerated, which can improve the exchange rate of materials between the inside and outside of the cell [36]. For microorganisms, the accelerated fluidity of the cell membrane can improve the uptake rate of external nutrients, which is more conducive to their growth and reproduction. Thus, in this study, LAB may be more diverse in medium-temperature regions than in low-temperature regions (Figure 4A,C–E). Compared with dry areas, humid areas can provide moisture to microorganisms to obtain the nutrient factors needed from the environment for growth [37]. Meanwhile, water can not only provide a place for biochemical reactions but can also participate in biochemical reactions [38,39]. Therefore, in this study, the species of LAB obtained from medium-humidity areas were more diverse than those obtained from low-humidity areas (Figure 4B–E). Compared with the medium-temperature and medium-humidity areas, the number of species of LAB was reduced in the high-temperature and high-humidity areas. Therefore, in addition to temperature and humidity, there may be other environmental factors that can affect the distribution of LAB among regions.

### 3.2. Genetic Diversity of L. plantarum

Not only was *L. plantarum* isolated from seven types of fermented foods (Figure 3), but its number was also the highest of all strains (Figure 1C). In addition, *L. plantarum* was the most widely distributed in 84 regions in Yunnan (Figure 4). Therefore, in this study, the whole genomes of *L. plantarum* (54 strains) from three different fermented food types were

analyzed. Among the 54 strains, 53 strains of *L. plantarum* were isolated by our laboratory, and the other 1 (NZ_CP009236.1) was obtained from the NCBI GenBank database. The details of the 54 strains are shown in Table S1.

### 3.2.1. General Genomic Features

The differences in the GC content and genome size of *L. plantarum* obtained from different sources are listed in Table 1. The results show that there was no significant difference in the GC content or genome size among the three groups ($p < 0.05$), indicating that natural selection in different ecological niches has resulted in structural variations in only a few core genes. This result is consistent with the finding of Pan et al. [12].

As shown in Figure S2, the GC content fluctuates a little, while the genome size fluctuates greatly. The average GC content of the 54 strains of *L. plantarum* was 44.28%, ranging from 35.60% to 47.90% (Table 1 and Figure S2A). The average genome size was 3.59 Mb, ranging from 2.54 Mb to 5.76 Mb (Table 1 and Figure S2B). The GC content and genome size were considered to be closely related to bacterial genome evolution and habitat adaptation [40]. Recent studies have shown that nomadic bacteria have a higher GC content and genome size than symbiotic bacteria [30]. *L. plantarum* is a typical nomadic bacterium, and *Lactobacillus reuteri* is a relatively common symbiotic bacterium [18,41]. The GC content and genome size of *L. reuteri* are typically around 38.6% and 2.28 Mb, respectively [41]. In this study, the average GC content and average genome size of *L. plantarum* were 44.28% and 3.59 Mb (Table 1), respectively, which were significantly higher than those of *L. reuteri*. This may be due to the stable environment provided by the host, rendering functions that were essential in the free-living ancestor superfluous, leading to an accumulation of loss-of-function mutations and pseudogenes followed by the removal of these genetic regions [42]. In addition, the GC content was considered to be closely related to energy metabolism. A high GC content in genomes may lead to an increased energy consumption of microorganisms during reproduction, whereas a low GC content in genomes may reduce energy consumption and maintain genomic stability [43]. Moreover, genome increases were strongly correlated with the environmental adaptation of LAB species, with the genome size being significantly higher in the nomadic and free-living species than in the host-adapted species [30]. Compared to *L. reuteri*, *L. plantarum* efficiently migrates to different habitats. Therefore, the genome size and GC content of *L. plantarum* were larger than those of host-adapted lactobacilli, allowing a wider complement of functional genes, thereby providing adaptive advantages in various habitats [30].

**Table 1.** Analysis of significant differences in GC content and genome size of *Lactobacillus plantarum* isolated from different sources.

| | Fermented Vegetables ($n = 37$) | Fermented Bean Products ($n = 9$) | Fermented Meat Products ($n = 8$) | Average |
|---|---|---|---|---|
| Content of GC (%) | 44.21 ± 0.27 [a] | 44.68 ± 0.34 [a] | 44.16 ± 0.05 [a] | 44.28 ± 0.20 |
| Genome size (Mb) | 3.63 ± 0.10 [a] | 3.63 ± 0.22 [a] | 3.37 ± 0.02 [a] | 3.59 ± 0.08 |

Different lowercase letter means the significant difference ($p < 0.05$); same lowercase letter means not significant difference ($p > 0.05$).

### 3.2.2. ANI Analyses of *L. plantarum*

ANI analyses have often been used to distinguish species relationships between strains [44]. In existing studies, ANI > 62% has been identified as the threshold boundary of the same genus, and ANI > 95% has been identified as the threshold boundary of the same species [45,46]. The ANI of the *L. plantarum* strains was analyzed to further investigate their uniqueness or potential subspecies. The ANI values of all strains, except for C735, E262 and E554, were higher than 96% (Figure 5A), indicating that these 51 strains belonged to the same species. The ANI values of C735, E262 and E554 were < 95% and >62%, suggesting

that these three strains may be potential subspecies of *L. plantarum*. However, further studies are needed to confirm this finding.

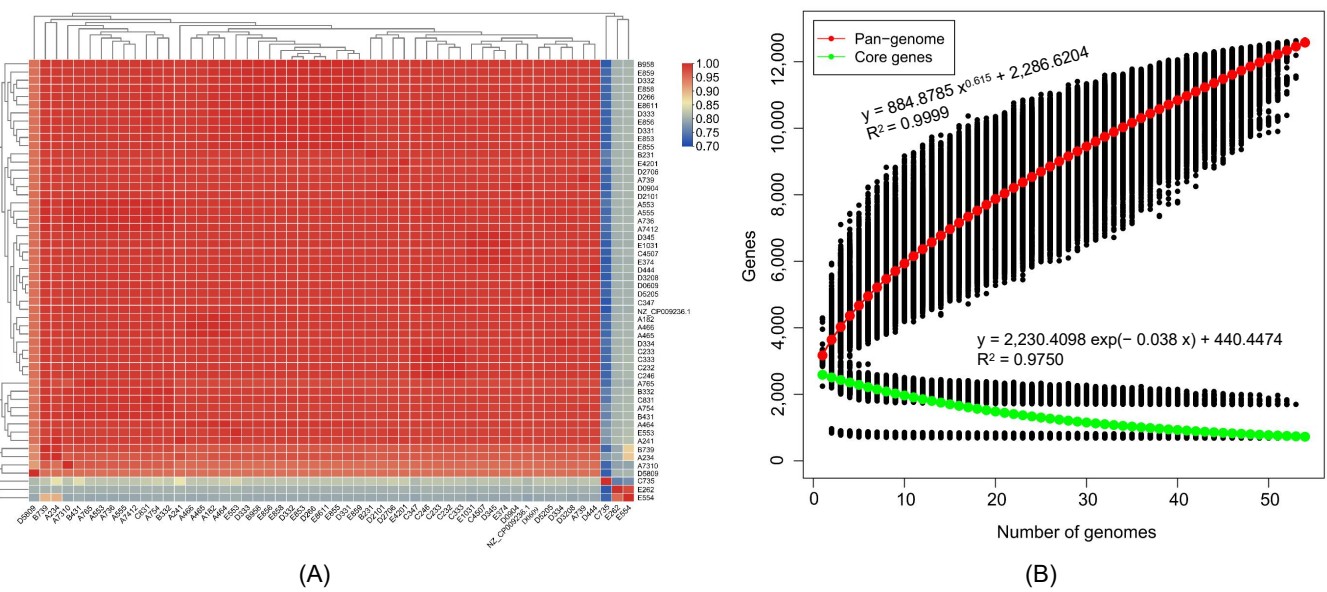

(A)                                                                                              (B)

**Figure 5.** ANI, pan-genome and core gene analyses of *Lactobacillus plantarum*. (**A**) ANI heatmap of 54 strains of *L. plantarum*; (**B**) pan-genome and core gene trend map of 54 strains of *L. plantarum*.

### 3.2.3. Pan-Genome and Core Genes of *L. plantarum*

The pan-genome is considered to be the entire genome of a species, including the core genes and variable genes [47]. Previous studies have reported that the pan-genome is helpful in exploring the evolution and genetic information of a species [48]. Thus, the pan-genome and core-genome trends of *L. plantarum* were predicted using PAGP. The result shows that a number of *L. plantarum* had a certain relationship with the number of pan-genes and core genes (Figure 5B). The 54 strains of *L. plantarum* had a total of 12,656 pan-genes and 688 core genes. The pan-genome trend graph shows that, as the number of *L. plantarum* genomes increased, the size of the pan-genome increased correspondingly, while the number of core genes gradually stabilized (Figure 5B). The exponential value of the derived mathematical equation was > 0.5, indicating that the pan-genome of *L. plantarum* was in an open state, as well as reflecting the huge number of pan-genomes of *L. plantarum* [18]. The ecological niche of *L. plantarum* was widely distributed; therefore, it can both exchange various genetic materials with and constantly acquire new genes from the outside world [18]. Thus, the pan-genome of *L. plantarum* is large. Additionally, the pan-genome of *L. plantarum* was in an open state, which also illustrates the diversity of the genomes and the complexity of the evolution of *L. plantarum*.

### 3.2.4. Homologous Genes and Phylogenetic Analyses

In modern molecular biology, homology refers to the similarity between genes [49]. A homology analysis of *L. plantarum* showed that there were 1150 homologous genes in the 54 strains (Figure 6A), indicating that 54 strains of LAB had a high homology. Unique genes refer to genes that only exist in specific strains [15]. The number of unique genes of the 54 strains of *L. plantarum* fluctuated between 2 and 1206 (Figure 6A), indicating that the strain has genetic diversity.

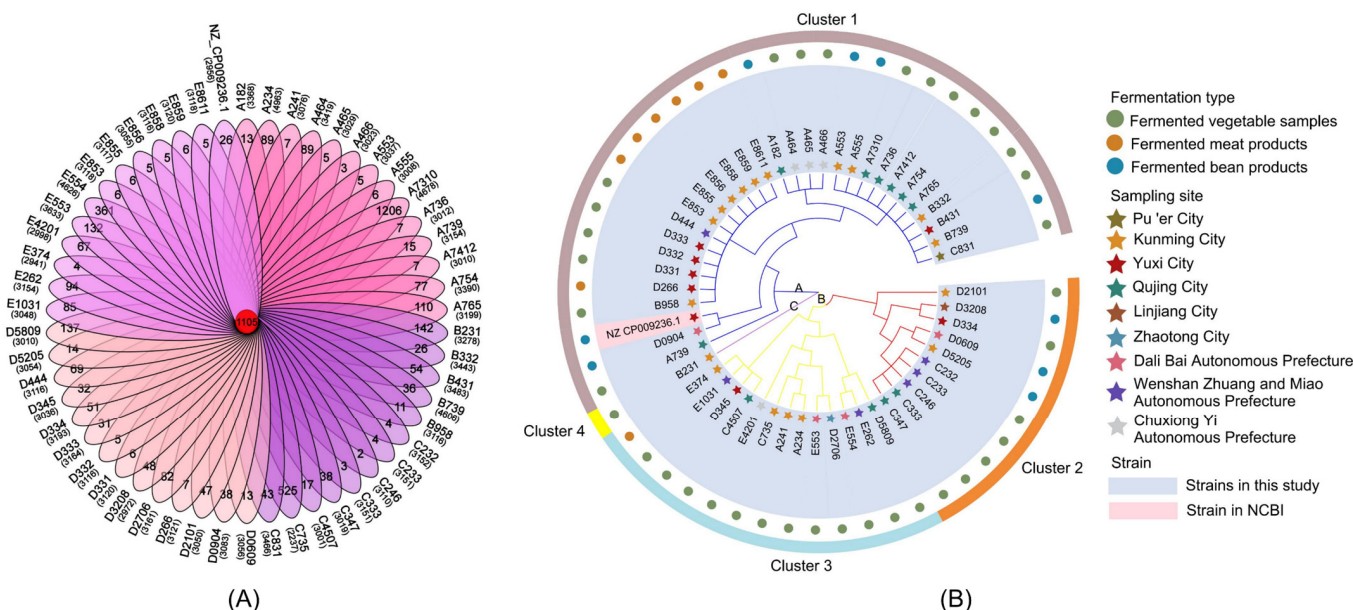

**Figure 6.** Homology analysis and phylogenetic analysis of *Lactobacillus plantarum*. (**A**) Venn diagram presenting the homology analysis of 54 strains of *L. plantarum*; (**B**) phylogenetic tree construction of 54 strains of *L. plantarum* based on homologous genes.

A phylogenetic tree of *L. plantarum* was constructed based on the homologous genes to investigate the phylogenetic relationships (Figure 6B). The 54 strains were grouped into three main branches and then into four clusters. The clustering results are essentially consistent with the ANI analyses, and the strains with lower ANI values were all clustered in branch B. Furthermore, *L. plantarum* from the same habitat source and geographic origin did not exist in the same branch, indicating that the homologous genes of *L. plantarum* are not related to the source of isolation (Figure 6B). This result is consistent with the recent findings of Martino et al. [18]. Previous research has shown that *L. plantarum* had the ability to migrate across environments [30], but the study conducted by Martino et al. showed that the genomic adaptation of *L. plantarum* may be driven by selective pressures rather than specific environmental adaptation [18].

### 3.2.5. Functional Annotation Analysis of COG Database

The COG database was used to annotate the genomes of *L. plantarum*, and the distributions of the functional genes in the genome were explored (Figure 7). Among the information storage and processing groups, transcription (K) genes were the most abundant, accounting for 9.72% of the total genes. In the category of cellular processes and signaling, cell wall/membrane/envelope biogenesis (M) had the most genes, accounting for 5.39% of the total genes. The carbohydrate transport and metabolism (G) and amino acid transport and metabolism (E) genes were the most frequent in carbohydrates, accounting for 8.36% and 7.11% of the total genes, respectively. However, there were a large number of genes with unknown functions in the genome of *L. plantarum*, indicating that the current research on the gene function of *L. plantarum* is not enough. The A, B, W, Y and Z genes were not present in the genome of *L. plantarum*. *L. plantarum* is a prokaryote; therefore, its genome obviously lacked any genes encoding RNA processing and modification, chromatin structures and dynamics, extracellular structures and nuclear structures [50].

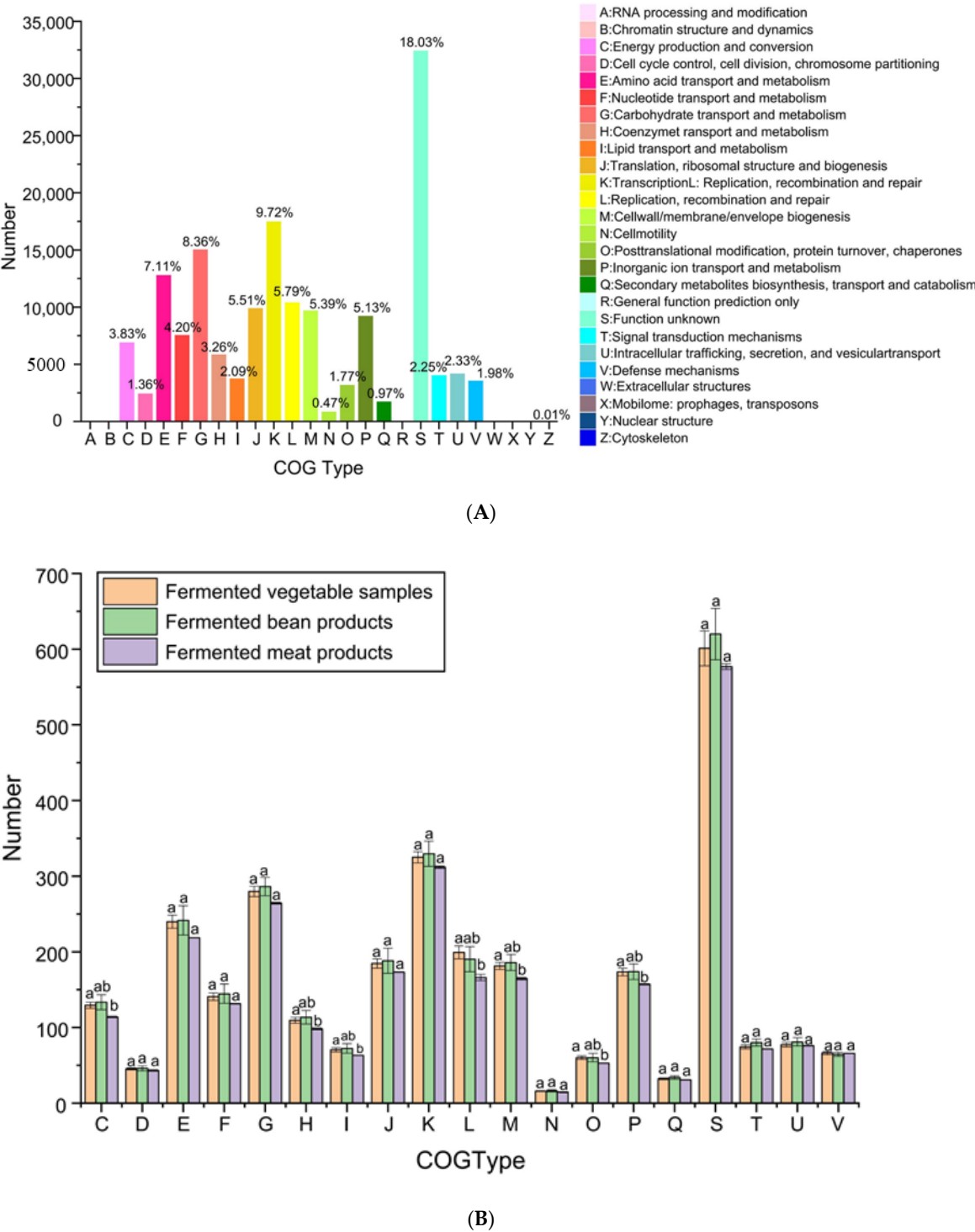

**Figure 7.** Analysis of COG functional annotation. (**A**) COG gene distribution in 54 strains of *Lactobacillus plantarum*; (**B**) the number of COG gene distributions of *L. plantarum* isolated from different sources. Different lowercase letter means the significant difference ($p < 0.05$); same lowercase letter means not significant difference ($p > 0.05$).

Figure 7B further reveals that *L. plantarum* selected from the fermented vegetables and fermented meat products showed differences in energy production and transformation (C); coenzyme transport and metabolism (H); lipid transport and metabolism (I); replication, reorganization and repair (L); cell wall/membrane/envelope biogenesis (M); post-translational modification, protein turnover and chaperone (O); and inorganic ion

transport and metabolism (P) ($p < 0.05$). This result indicates that environment-specific selection pressures drive the adaptation of *L. plantarum* to specific ecological niches. Furthermore, previous research has shown that, while some genes are lost in the course of adapting to a particular environment, genes that are critical for survival or provide a competitive advantage are retained [15]. Therefore, the functional genes of *L. plantarum* in the fermented vegetables and fermented meat products showed discrepancies.

3.2.6. Carbohydrate Enzyme Analysis of *L. plantarum*

Figure 8A shows that glycoside hydrolase families (GHs), glycosyltransferase families (GTs), carbohydrate esterase families (CEs), carbohydrate-binding modules (CBMs) and auxotrophic active enzyme families (AAs) were present in the 54 strains of *L. plantarum*. Among the five families of carbohydrate-active enzymes, the enzyme-encoding genes of the GH and GT families were abundant in *L. plantarum*, while the enzyme-encoding genes of the CE, CBM and AA families were less abundant. In addition, the PL gene was not present in all strains of *L. plantarum*. Figure 8B further shows that there was no significant difference in the distribution of the AA, CBM, CE and GT genes in the three habitat groups ($p < 0.05$). However, the GH genes were different in the three habitat groups. Glycosidase (GH) was abundant in almost all living organisms, and it was primarily involved in the transglycosylation or hydrolysis of the glycosidic bonds present in glycosides, glycoconjugates and glycans [51]. A possible reason for this is that fermented vegetables and fermented bean products are richer in cellulose and other polysaccharides than fermented meat products [52,53]. Therefore, the number of GH genes in the fermented vegetables and fermented bean products was significantly higher than those in the fermented meat products ($p < 0.05$).

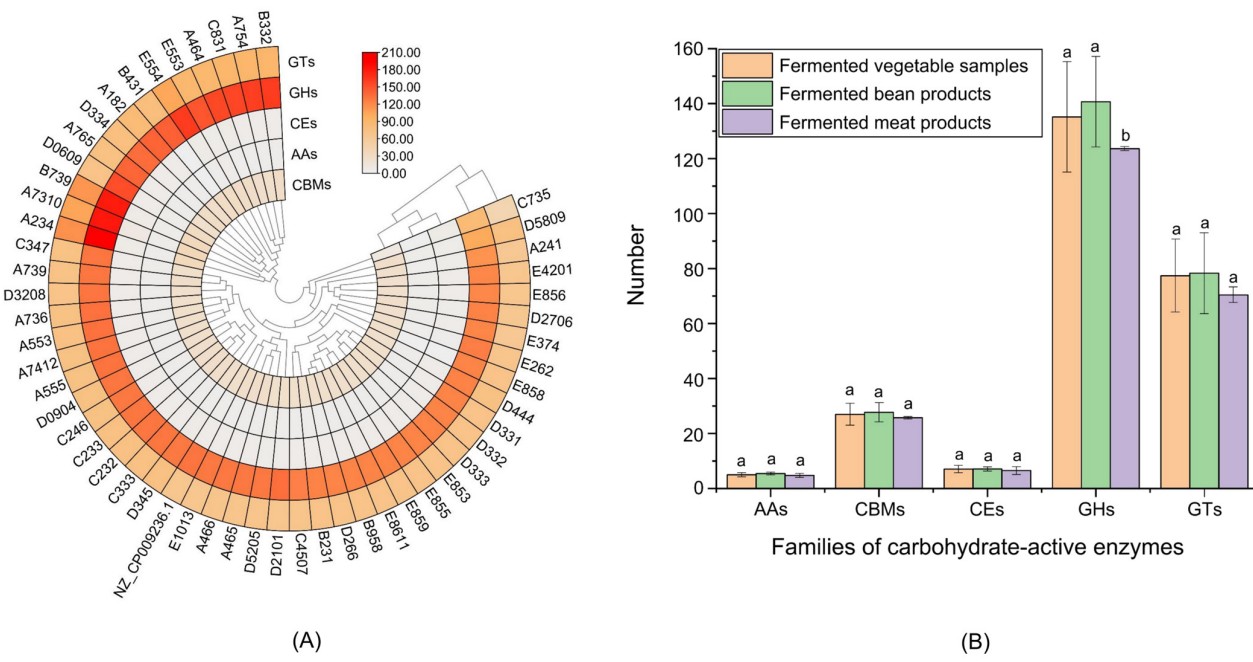

(A)  (B)

**Figure 8.** *Cont.*

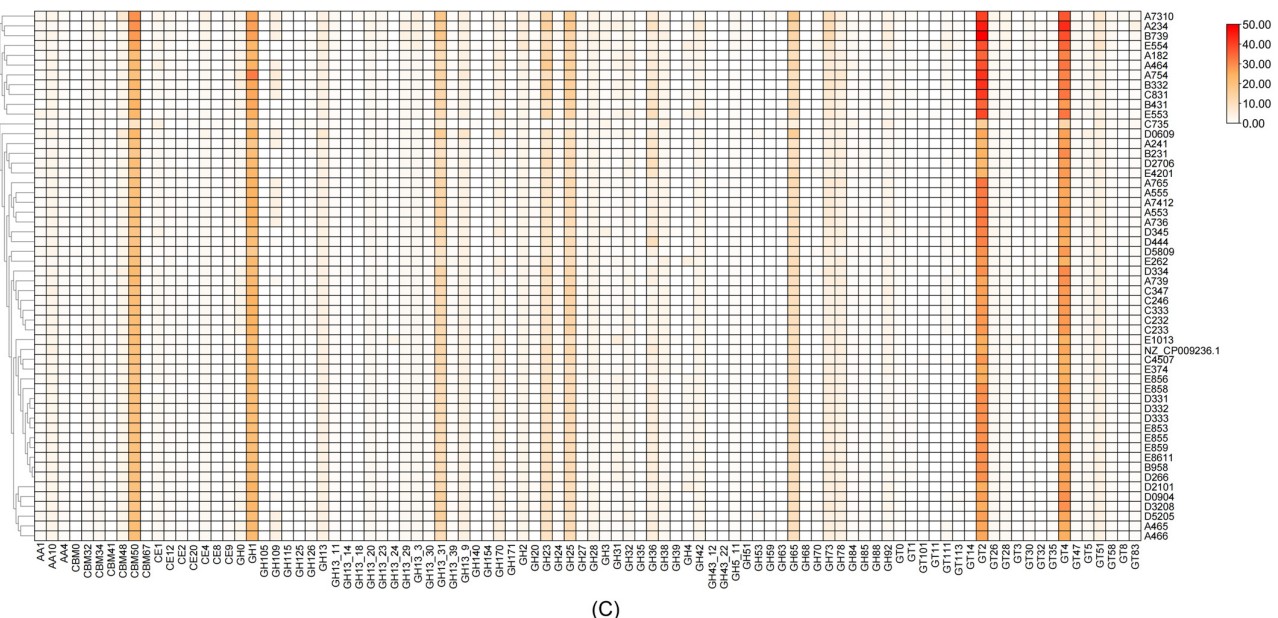

(C)

**Figure 8.** Genotype analysis of carbohydrate utilization in *Lactobacillus plantarum*. GHs: glycoside hydrolase families; GTs: glycosyltransferase families; CEs: carbohydrate esterase families; CBMs: carbohydrate-binding modules; AAs: auxotrophic active enzyme families. (**A**) Heatmap of gene abundance distribution of five families of carbohydrate-active enzymes in *L. plantarum*; (**B**) the number of carbohydrate-active enzymes in different types of fermented foods. Different lowercase letter means the significant difference ($p < 0.05$); same lowercase letter means not significant difference ($p > 0.05$); (**C**) prediction of carbohydrate-active enzyme of 54 strains of *L. plantarum*.

Figure 8C shows the analysis of the five major carbohydrate-active enzyme families, and the result shows that there were 95 carbohydrate-active enzyme-related genes in the 54 strains of *L. plantarum*. Among the 95 genes, GH1, GH13, GH23, GH24, GH25, GH27, GT4, GT2, GT26, GT51, GT58, CE4, CE9, AA10 and 26 other genes were shared by the 54 strains of *L. plantarum*. However, the distributions of the remaining 55 carbohydrate-active enzyme-related genes were different among these strains, indicating that the CAZY enzyme activities of the different *L. plantarum* strains had genetic diversity.

## 4. Conclusions

In this research, the diversity of LAB in six types of fermented foods collected from 84 regions in Yunnan was analyzed, and the genetic diversity of *L. plantarum* was also investigated. The results show that *Lactobacillus* was the dominant genus in the traditional fermented foods obtained from Yunnan. *L. plantarum* was the most abundant in the fermented foods. Furthermore, the result also shows that the dominant strains of the various fermented food types were different. *L. mesenteroides* and *E. faecium* were the dominant strains in fermented vegetables. *L. backii*, *L. pobuzihii* and *L. zhachilii* were the dominant strains in fermented bean products. *W. confuse* was the dominant strain in wine samples. *L. mesenteroides* was the dominant strain in fermented meat products. The dominant strains in fermented dairy products, fermented flour samples, fermented fruits and fermented tea were *W. paramesenteroides*, *W. viridescens*, *L. nagelii* and *P. acidilactici*, respectively. The majority of the LAB in the fermented foods obtained from Yunnan were better adapted to regions where the temperature is 15–20 °C and the humidity is 64–74%. In addition, *L. plantarum* was the most widely distributed in the 84 regions. The average GC content and average genome size of the 54 strains of *L. plantarum* were 44.28% and 3.59 Mb, respectively. Additionally, the pan-genome of *L. plantarum* was in an open state, indicating that its pan-genome was tremendous. A phylogenetic analysis revealed that the habitat source and geographic origin had little influence on the homologous genes of *L. plantarum*.

Furthermore, the genetic diversity of *L. plantarum* was mainly manifested in functional genes and carbohydrate utilization. In general, this research provides the distributions of culturable microbial species in fermented foods collected from Yunnan and the genomic characteristics of *L. plantarum*, laying the foundations for future research.

**Supplementary Materials:** The following supporting information can be downloaded at: https://www.mdpi.com/article/10.3390/fermentation9040402/s1, Figure S1: A pie chart of the number of samples of culturable lactic acid bacteria and the number of samples of non-culturable LAB; Figure S2: GC content and genomic size of *Lactobacillus plantarum*. (A) GC content distribution of *L. plantarum*. (B) Genomic size distribution of *L. plantarum*; Table S1: Detailed information of 54 strains of *Lactobacillus plantarum*.

**Author Contributions:** Conceptualization, J.Y.; methodology, Z.L. and C.C.; software, C.C. and H.L.; validation, J.Y.; formal analysis, J.Z.; investigation, H.L., J.Z. and Y.X.; resources, J.Y.; data curation, Y.S. and S.Z.; writing—original draft preparation, H.L.; writing—review and editing, C.C. and Z.L.; visualization, Y.S.; supervision, J.Y.; project administration, X.H.; funding acquisition, J.Y. All authors have read and agreed to the published version of the manuscript.

**Funding:** This research was funded by the Yunnan Provincial Natural Science Foundation (202101BE070001-054), the Excellent Youth Funding of Yunnan Province (YNQR-QNRC-2018-109), and the Young Elite Scientists Sponsorship Program of China Association for Science and Technology (YESS20200123).

**Institutional Review Board Statement:** Not applicable.

**Informed Consent Statement:** Not applicable.

**Data Availability Statement:** Not applicable.

**Conflicts of Interest:** The authors declare no conflict of interest.

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
