# Peer review of "Biodiversity of Lactic Acid Bacteria in Traditional Fermented Foods in Yunnan Province, China, and Comparative Genomics of Lactobacillus plantarum"

_fermentation, doi:10.3390/fermentation9040402_

Round 1
Reviewer 1 Report
1-Add China after Yunnan Province in the title of the Ms
2-Correct the format of Lactobacillus plantarum at Line 4.and L305, L318, L388
3-L165 complete the sentences (in different fermented?)
4-show the used abbreviations 8A and B
5-correct W confuse at L465
6-carefully check all references especially for scientific names of microorganisms see for example L472/481/485/496/498/526….etc
7-Carefully check all journal names that should be started with Capital letter see for example L528
Reviewer 2 Report
Fermentation 2290871
General
The manuscript deals with the biodiversity of lactic acid bacteria in traditional fermented foods in Yunnan Province and comparative genomics of Lactobacillus plantarum. Most of the species grow at 15-20ºC and 64-74% humidity. Habitat source and geographic origin scarcely influence the homologous genes while diversity was mainly due to functional genes and carbohydrate use. The research may increase information on the fermented Yunnan foods and understand the evolutionary history of the strain. The work is wide and intense. As general suggestions, please consider improving font sizes of legends in Figures, if possible, and correct wordy sentences (e.g., It can also be seen in Figure 4…., pg 6, line 230). Besides, bar graphs are not generally appropriate for informing on populations (average, dispersion, etc,), although is widely used in biology. What is the meaning of the part of the bar below the inferior limit of confidence? Use box-plots instead.
Specific suggestions.
L21. GC. Please full name first time.
L27. Species?
L15-27. Please, reduce wordy.
L41-43. Possibly wordy sentence. Please, be concrete.
L83. Please include NaCl content.
L95. Thee are several biodiversity indices. No one of them was useful?
L135-138. Was considered the multiple comparison effect?
L149, 151, …. through text. A total… Necessary?
Fig. 1. Could enlarge legends to make them more legible? Maybe there is margin to do it without distorting the graph. Please, revise the other Fig. with the same aim.
L318, 388… Use italic. Please, revise.
Fig. 8. In my opinion, the use of intense blue tone for absence of predictions may be confusing. Usually, intense tones are associated with high presence and, maybe, reduce relevance to those cells with predictions.
L426 Italic. Please, revise.
L 430 and throughout text. The use of prefer for this context should be revised (e.g., adapted) How do you measure preference (a sensation)?
441-442. The titles of the supplementary material are missing.
